# Harmful DNA:RNA hybrids are formed *in cis* and in a Rad51-independent manner

**Juan Lafuente-Barquero, Maria Luisa García-Rubio, Marta San Martin-Alonso, Belén Gómez-González\*, Andrés Aguilera\***

Centro Andaluz de Biología Molecular y Medicina Regenerativa-CABIMER, Universidad de Sevilla-CSIC-Universidad Pablo de Olavide, Seville, Spain

**Abstract** DNA:RNA hybrids constitute a well-known source of recombinogenic DNA damage. The current literature is in agreement with DNA:RNA hybrids being produced co-transcriptionally by the invasion of the nascent RNA molecule produced in cis with its DNA template. However, it has also been suggested that recombinogenic DNA:RNA hybrids could be facilitated by the invasion of RNA molecules produced in trans in a Rad51-mediated reaction. Here, we tested the possibility that such DNA:RNA hybrids constitute a source of recombinogenic DNA damage taking advantage of Rad51-independent single-strand annealing (SSA) assays in the yeast *Saccharomyces cerevisiae*. For this, we used new constructs designed to induce expression of mRNA transcripts in trans with respect to the SSA system. We show that unscheduled and recombinogenic DNA:RNA hybrids that trigger the SSA event are formed in cis during transcription and in a Rad51-independent manner. We found no evidence that such hybrids form in trans and in a Rad51-dependent manner.

**\*For correspondence:**
gomezb@us.es (BéGó-Gá);
andres.aguilera@cabimer.es (AéA)

## Introduction

R loops are structures formed by a DNA:RNA hybrid and the complementary displaced single stranded DNA (ssDNA). They were observed naturally as programmed events in specific genomic sites such as the S regions of Immunoglobulin genes in mammals or mitochondrial DNA (*Chang et al., 1985*; *García-Muse and Aguilera, 2019*; *Yu et al., 2003*), where they play specific functions by promoting class switch recombination or DNA replication, respectively; but also as unscheduled non-programmed structures upon dysfunction of RNA binding proteins involved in the assembly or processing and export of the protein-mRNA particle (mRNP) such as the THO complex or the SRSF1 splicing factor (*Huertas and Aguilera, 2003*; *Li and Manley, 2005*). Also, they have been inferred in the rDNA regions of the bacterial chromosome upon Topo I inactivation (*Drolet et al., 1995*). Accumulated evidence indicates that R loops are detected from yeast to humans in many transcribed regions of the eukaryotic genome in wild-type cells, in cells defective in several metabolic processes covering from RNA processing to DNA replication and repair and in cells deficient in specific chromatin factors (*Bhatia et al., 2014*; *García-Muse and Aguilera, 2019*; *García-Rubio et al., 2015*; *Herrera-Moyano et al., 2014*; *Mischo et al., 2011*; *Paulsen et al., 2009*; *Schwab et al., 2015*). The biological consequences of such R loop structures are diverse and include replication stress, DNA breaks and genome instability that can be detected as hyperrecombination, plasmid loss or gross chromosomal rearrangements (*García-Muse and Aguilera, 2019*). Indeed, DNA:RNA hybrids have been inferred by their potential to induce DNA damage and recombination, but they can also be directly detected via different methodologies. These include electrophoresis detection after nuclease treatment, bisulfite mutagenesis or either in situ immunofluorescence or DNA:RNA Immuno-Precipitation (DRIP) using the S9.6 anti-DNA:RNA monoclonal antibody (*García-Muse and Aguilera, 2019*).

The increasing number of reports showing R loop accumulation in different organisms from bacteria to human cells, and the relevance of their functional consequences, whether on genome integrity, chromatin structure and gene expression suggest that most DNA:RNA hybrids are compatible with a co-transcriptional formation (*García-Muse and Aguilera, 2019*). This is consistent with the idea that it is the RNA produced in cis the one that invades the duplex DNA, a reaction that can be facilitated by DNA sequence and supercoiling (*Stolz et al., 2019*) as well as by nicking of the DNA template (*Roy et al., 2010*). The evidence of DNA:RNA hybrid formation at breaks has matured in the last years (*Cohen et al., 2018*; *D'Alessandro et al., 2018*; *Li et al., 2016*; *Ohle et al., 2016*; *Teng et al., 2018*; *Yasuhara et al., 2018*) although the source and role of such hybrids remains still controversial (*Aguilera and Gómez-González, 2017*; *Puget et al., 2019*). Of note, genome-wide mapping results have been interpreted in diverse manners by different labs. Whereas some claim that DNA:RNA hybrids detected around DNA breaks mostly accumulate at transcribing sites (*Cohen et al., 2018*), in agreement with their co-transcriptional formation, others suggest that there is no preference for DNA:RNA hybrids to form at transcribed loci in human cells (*D'Alessandro et al., 2018*), implying a scenario in which DNA:RNA hybrids at break sites would form either de novo or with RNAs produced at different loci (in trans). Moreover, it has been shown in yeast that short RNAs can be used as templates for the recombinational repair of DSBs in a reaction catalyzed by Rad52 (*Keskin et al., 2014*).

DNA:RNA hybrids can also form in vitro with the aid of the bacterial DNA strand exchange protein RecA (*Kasahara et al., 2000*; *Zaitsev and Kowalczykowski, 2000*). In vivo, DNA:RNA hybrids are formed with RNAs produced in trans as intermediates in the course of ribonucleoprotein-mediated reactions such as telomerase and CRISPR-Cas9 ribonucleoprotein involved in specific reactions (*Collins, 2000*; *Jinek et al., 2012*). They have also been reported to have regulatory roles in gene expression when formed by long non-coding RNAs (lncRNAs) at in trans loci such as the cases of the GAL lncRNA in yeast (*Cloutier et al., 2016*) or the APOLO lncRNA in plants (*Ariel et al., 2020*). In summary, despite the accumulating evidence that in vivo DNA:RNA hybrids formed in cis constitute a threat for genome stability, an open question is whether DNA:RNA hybrids also form in trans as a potential source of recombinogenic DNA damage. To our knowledge, this has only been addressed in the yeast *Saccharomyces cerevisiae* (*Wahba et al., 2013*). By S9.6 immunofluorescence (IF) and a yeast artificial chromosome-based genetic assay that measures gross chromosomal rearrangements, it was inferred that DNA:RNA hybrids could be formed with RNAs produced in trans by a reaction catalyzed by the eukaryotic DNA strand exchange protein Rad51 (*Wahba et al., 2013*). Nevertheless, the fact that the detected gross chromosomal rearrangements could depend on Rad51 and that the S9.6 antibody can also recognize dsRNAs (*Hartono et al., 2018*; *König et al., 2017*; *Silva et al., 2018*), prompted us to address this question using a different approach. Using Rad51-independent recombination assays in which the initiation region could be unambiguously delimited, we do not find evidence for recombinogenic DNA:RNA hybrids forming in trans. Instead, we provide genetic evidence that DNA:RNA hybrids compromising genome integrity are formed in cis and in a Rad51-independent manner.

## Results

### A new genetic assay to detect recombinogenic DNA:RNA hybrids with RNA produced in trans

We developed a new genetic assay to infer the formation of recombinogenic DNA:RNA hybrids with RNAs produced in trans. It is based on two plasmids, one containing the recombination system and the *LacZ* gene in cis (GL-*LacZ* recombination system), and another one providing the in trans LacZ transcripts (*tet$_p$:LacZ*) (*Figure 1*). The bacterial *LacZ* gene consists of a 3 Kb sequence with high G+C content previously reported to be hyper-recombinant and difficult to transcribe in DNA:RNA hybrid-accumulating strains, such as *tho* mutants (*Chávez et al., 2001*).

The GL-*LacZ* recombination system is a *leu2* direct-repeat construct carrying the *LacZ* gene in between and under the *GAL1* inducible promoter so that this construct is transcribed as a single RNA unit driven from the *GAL1* promoter (*Piruat and Aguilera, 1998*). Single-Strand Annealing (SSA) events cause the deletion of the *LacZ* sequence and one of the *leu2* repeats leading to Leu+ recombinants in a Rad51-independent manner (*Figure 1A*). To provide *LacZ* transcripts in trans, we

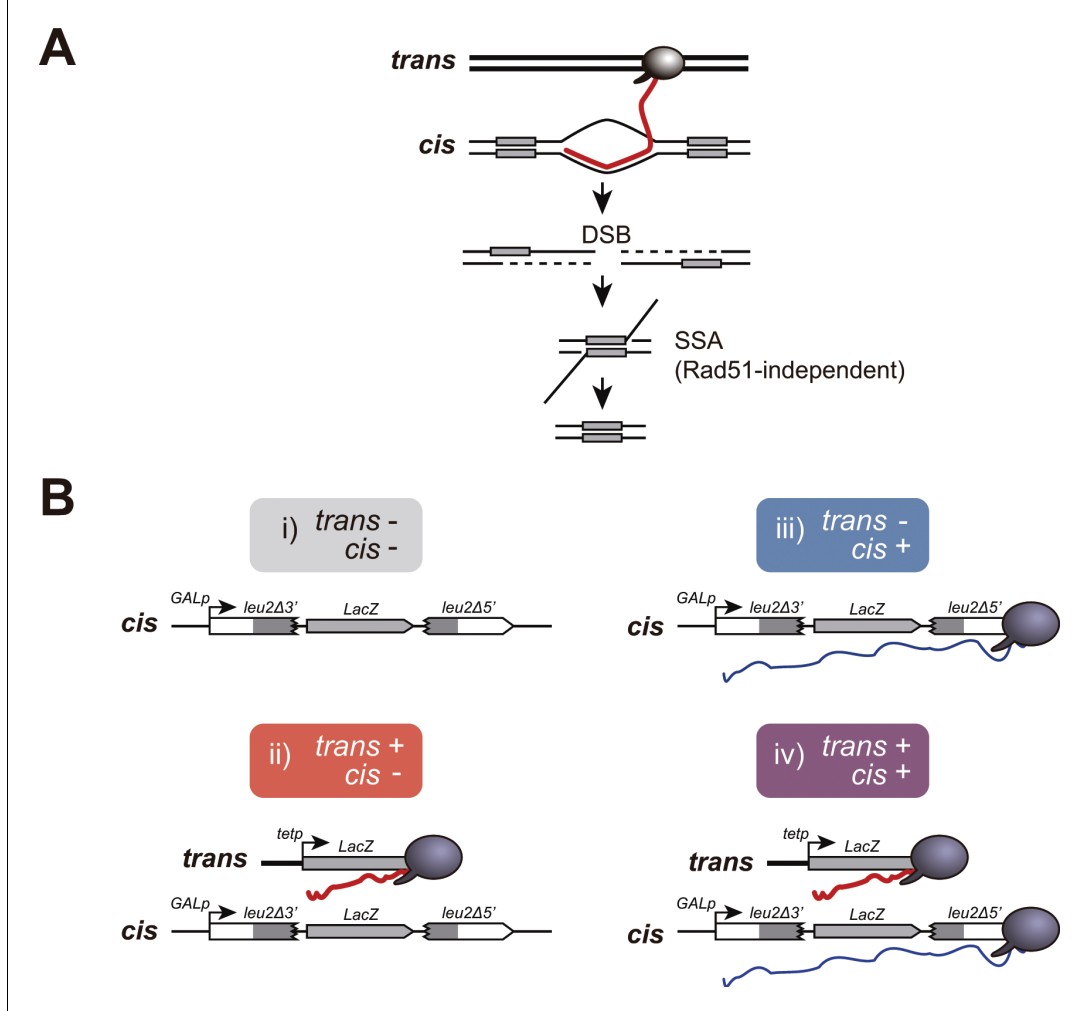

**Figure 1.** A new genetic assay to detect recombinogenic DNA:RNA hybrids in trans. (**A**) DSBs induced in between direct repeats by DNA:RNA hybrids putatively formed with RNA produced in trans would be repaired by Rad51-independent Single-Strand Annealing (SSA) causing the deletion of one of the repeats. A DSB is depicted for simplicity, but other recombinogenic lesions such as nicks or ssDNA gaps cannot be ruled out. (**B**) Schematic representation of the recombination assay to study the recombinogenic potential RNA produced by transcription (Trx) in cis or in trans. Four combinations were studied: i) no transcription, with GL-*LacZ* construct turned transcriptionally off (2% glucose) and an empty plasmid; ii) transcription in trans, with GL-*LacZ* construct turned transcriptionally off (2% glucose) and the *tetₚ:LacZ* construct; iii) transcription in cis, with GL-*LacZ* construct turned transcriptionally on (2% galactose) and an empty plasmid; and iv) transcription in cis and in trans, with GL-*LacZ* construct turned transcriptionally on (2% galactose) and the *tetₚ:LacZ* construct.

The online version of this article includes the following source data and figure supplement(s) for figure 1:

**Figure supplement 1.** *LacZ* expression levels in the GL-*LacZ* and *tetₚ:LacZ* constructs.
**Figure supplement 1—source data 1.** Relative RNA levels at theLacZgene from GL-LacZandtetp:LacZconstructs.

used a fusion construct containing the complete bacterial *LacZ* gene sequence under the doxycycline-inducible *tet* promoter (*tetₚ:LacZ*). As a control of no expression in trans, we used transformants with an empty plasmid to avoid any possible effect from leaky transcription from the *tet* promoter in the presence of doxycycline.

Yeast strains carrying both GL-*LacZ* recombination system and the *tetₚ:LacZ* construct were used to assay SSA events in the four different possible conditions: i) no transcription, with GL-*LacZ* construct turned transcriptionally off (2% glucose) and an empty plasmid; ii) transcription in trans, with GL-*LacZ* construct turned transcriptionally off (2% glucose) and the *tetₚ:LacZ* construct; iii) transcription in cis, with GL-*LacZ* construct turned transcriptionally on (2% galactose) and an empty plasmid;

and iv) transcription in cis and in trans, with GL-*LacZ* construct turned transcriptionally on (2% galactose) and the *tet$_p$:LacZ* construct (*Figure 1B*).

## RNAs produced in trans are not a spontaneous source of recombinogenic DNA damage

The analysis of recombination in wild-type cells revealed that whereas the stimulation of transcription in cis elevated the frequency of recombination threefold, the stimulation of transcription in trans driven from the *tet$_p$:LacZ* construct had no effect on recombination (*Figure 2A*). These results already suggest that homologous transcripts coming from a different locus do not represent a detectable source of genetic instability in wild-type conditions and thus argue against the hypothesis that spontaneous DNA:RNA hybrids could be formed with mRNAs generated in trans. However, it is known that mRNA coating protects DNA from co-transcriptional RNA hybridization. Thus, we wondered if transcripts produced in trans could induce recombination in mRNP-defective mutants such as those of the THO complex. Hence, we performed our experiments in *mft1Δ* and *hpr1Δ* mutant strains. *mft1Δ* and *hpr1Δ* enhanced recombination slightly when transcription in cis was switched off (*Figure 2A*), likely as a consequence of leaky transcription form the *GAL1* promoter in glucose (*Figure 1—figure supplement 1*). More significantly and in agreement with previous reports (*Chávez et al., 2000*), recombination frequencies rocketed when transcription was stimulated in cis. However, transcription activation in trans did not enhance recombination, as it would be expected if additional DNA:RNA hybrids could form with RNA produced in trans.

Instead, under conditions of high transcription of the recombination system (transcription in cis), RNA driven from an ectopic locus (transcription in trans) led to a partial suppression of the hyper-recombination. The reason for such suppression might involve the potential ability of the remotely produced RNAs to interfere with transcription occurring at the GL-*LacZ* construct. Given that a DNA:RNA hybrid produced in the template DNA strand can impair transcription elongation (*Tous and Aguilera, 2007*), one possibility would be that this interference is mediated by DNA:RNA hybrids formed between the RNA produced in trans and the transcribed DNA strand of the GL-*LacZ* construct. To rule out this possibility, we used an alternative recombination system (GL-*LacZi*), in which the *LacZ* sequence was inverted so that the *LacZ* transcript produced in trans would not be able to anneal with the transcribed DNA strand of the GL-*LacZi* system (*Figure 2B*). We detected a strong hyper-recombination in *hpr1Δ* cells when the *LacZ* sequence was transcribed in agreement with previous reports and with the fact that it has been shown that it is the length (and the GC content) but not the orientation of the *lacZ* sequence what impairs transcription and triggers hyper-recombination (*Chávez and Aguilera, 1997*; *Chávez et al., 2001*). Surprisingly, the production of RNAs in trans from the *tet::LacZ* construct also led to a reduction of the hyper-recombination in this system. Furthermore, in this case, the suppression was stronger and was also observed in glucose, when transcription in cis was off. This could be explained because, in this scenario, the RNA produced in trans is complementary to the mRNA produced in cis. Consequently, they can hybridize together forming a dsRNA that would preclude the possibility to form DNA:RNA hybrids at the GL-*LacZi* construct.

Since transcription from the long *LacZ* gene is inefficient and leads to unstable RNA products, particularly in *tho* mutants (*Chávez et al., 2001*), we made a new construct with only the last 400 bp of *LacZ* (*tet$_p$:LacZ400*) (*Figure 2C*). Strikingly, in this case, we observed no suppression of the *tho*-induced hyper-recombination by the production of RNA in trans. More importantly and again, recombination frequencies were not significantly enhanced by transcription in trans in any of the strains or conditions tested, further arguing against mRNA produced in trans as a possible source of recombinogenic DNA:RNA hybrids.

The THO complex is thought to prevent R-loops mainly by promoting a proper mRNA-protein assembly (*Luna et al., 2019*), whereas the two RNase H enzymes efficiently degrade the RNA moiety of DNA:RNA hybrids once formed (*Cerritelli and Crouch, 2009*). Thus, to favor DNA:RNA hybrid accumulation, we used cells lacking both RNases H1 and H2 and we determined the impact on SSA. *Figure 2A and C* show that *rnh1Δ rnh201Δ* cells elevated the recombination frequency when transcription was stimulated in cis, as expected. Importantly, the recombination frequencies were not altered by producing transcripts in trans, arguing again against the recombinogenic potential of putative DNA:RNA hybrids formed with RNAs produced in trans.

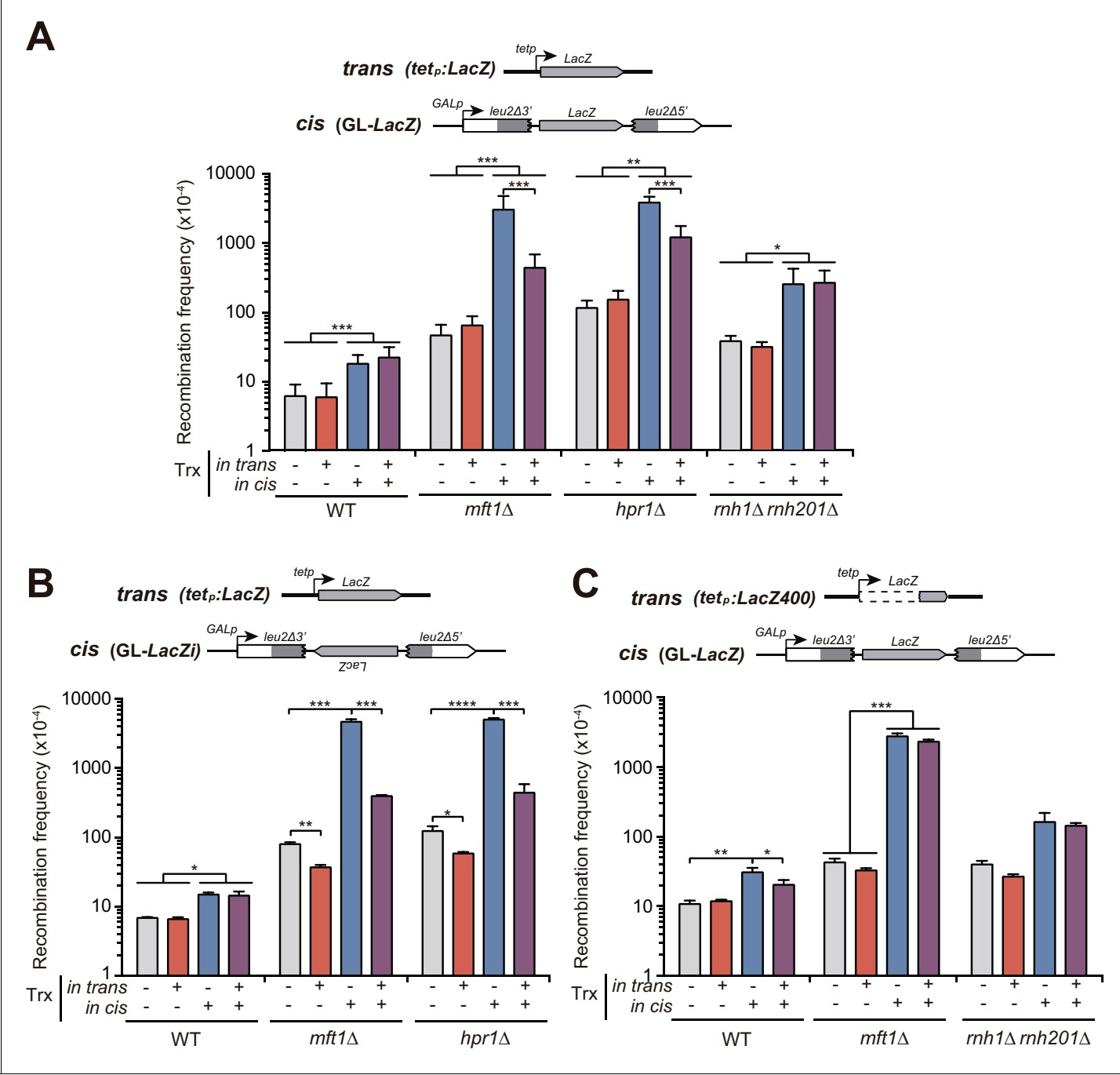

**Figure 2.** Analysis of the effect on genetic recombination of RNA produced in cis or in trans. (A) Recombination analysis in WT (W303), *rnh1Δ rnh201Δ* (HRN2.10C), *mft1Δ* (WMK.1A) and *hpr1Δ* (U678.4C) strains carrying GL-*LacZ* plasmid system (pRS314-GL-*LacZ*) plus either the pCM190 empty vector or the same vector carrying the *LacZ* gene (pCM179). (B) Recombination analysis in WT (W303), *mft1Δ* (WMK.1A) and *hpr1Δ* (U678.4C) strains carrying GL-*LacZi* plasmid plus either the pCM190 empty vector or the same vector carrying the sequence of the *LacZ* gene (pCM179). (C) Recombination analysis in WT (W303), *rnh1Δ rnh201Δ* (HRN2.10C) and *mft1Δ* (WMK.1A) strains carrying GL-*LacZ* plasmid system (pRS314-GL-*LacZ*) plus either the pCM190 empty vector or the same vector carrying the last 400 bp from the 3' end of the *LacZ* gene (pCM190:*LacZ400*). In all panels, average and SEM of at least three independent experiments consisting in the median value of six independent colonies each are shown. *, $p \leq 0.05$; **, $p \leq 0.01$; ***, $p \leq 0.001$; ****, $p \leq 0.0001$ (unpaired Student's t-test).

The online version of this article includes the following source data for figure 2:

**Source data 1.** Analysis of the effect on genetic recombination of RNA producedin cisorin trans.

In order to confirm DNA:RNA hybrid formation in these different sequence contexts, we performed DRIP experiments at the *LacZ* sequence (*Figure 3*) and we observed that DNA:RNA hybrids accumulate in both, *hpr1Δ* and *rnh1Δ rnh201Δ* mutants, and in all *GL-LacZ, tet:LacZ* and *GL:lacZi* sequences as expected, despite the technical difficulty of detecting increases in hybrid accumulation in plasmids, since they cause plasmid loss.

Given that the levels of transcription from the *GAL1* and *tet* promoters used for the constructs are very different (*Figure 1—figure supplement 1*), we decided to perform recombination tests with similar constructs in which the promoters were interchanged. Thus, we studied recombination in the TL-*LacZ* recombination system (*Santos-Pereira et al., 2013*) and used a *GAL:LacZ* construct to produce the *LacZ* transcripts from a remote locus. *Figure 4* shows that, whereas transcription at the TL-*LacZ* recombination system enhanced recombination as previously published (*Santos-Pereira et al., 2013*), again no significant stimulation of recombination was detected when RNAs were produced in trans in either wild-type, *hpr1Δ* or *rnh1Δ rnh201Δ* cells even when the RNA was generated from the strong *GAL1* promoter.

Finally, since all experiments were performed in plasmid-born systems in the original W303 background bearing the *rad5-G535R* mutation (*Fan et al., 1996*), we integrated the GL-*LacZ* system in the chromosome of a *RAD5* wild-type strain to ascertain that the *rad5-G535R* mutation did not affect the results as well as to confirm that the results were the same in a chromosome *locus*. As it can be seen in *Figure 5*, transcription of the chromosomal recombination system promoted a 30-fold increase in recombination levels in the *tho* mutant *hpr1Δ* with respect to the WT, in agreement with all previous data showing that co-transcriptional DNA:RNA hybrids are a potent source of recombination. By contrast, mRNA produced at a different locus had no effect on recombination, neither in wild-type cells nor in the *tho* mutant *hpr1Δ*. Hence, altogether, these results argue that, in contrast to mRNA produced in cis, RNA produced at a particular *locus* does not lead to recombinogenic DNA damage at regions located in trans.

## Rad51 is not required for DNA:RNA hybridization

We next wondered about the possible role of the recombination protein Rad51 in DNA:RNA hybridization. To examine this, we analyzed in *hpr1Δ* cells the effect of transcribing the ectopic *tet:LacZ* construct on recombination in our direct-repeat systems when these were not transcribed (*Figure 6*). It is important to remark that the recombination events detected in our assays are deletions occurring by SSA between direct repeats, which do not require Rad51 (*Pardo et al., 2009*). Indeed, in agreement with SSA annealing being Rad51-independent, *RAD51* deletion caused no significant changes in the recombination frequencies in our assay. Thus, any conclusion about Rad51-dependency or independency of the hybridization inferred from our assay is not contaminated by a possible direct role of Rad51 in the event we are studying. Importantly, we observed no differences when *RAD51* was deleted in *hpr1Δ* cells even when the *LacZ* sequence was expressed from the plasmid containing the *tet::LacZ* construct. This result argues against Rad51 facilitating or impeding the formation of DNA:RNA hybrids with RNAs produced in trans.

We then wondered whether the formation of known recombinogenic DNA:RNA hybrids formed in cis, such as those reported in the *hpr1Δ* mutant, requires Rad51. For this purpose, we studied the effect in the strong hyper-recombination phenotype of *hpr1Δ* when transcription was induced in cis. As shown in *Figure 6*, the absence of Rad51 had no effect on the hyper-recombination observed, as *hpr1Δ rad51Δ* cells elevated the recombination frequency more than 70-fold with respect to *rad51Δ*, similarly to Rad51+ cells. This result clearly indicates that the in cis DNA:RNA hybrid-mediated hyper-recombination phenotype is actually independent on Rad51.

In parallel, we studied the formation of Rad52 foci, a marker of recombinogenic DNA breaks (*Lisby et al., 2001*), in which case we used AID overexpression to enhance the recombinogenic potential of R loops (*Gómez-González and Aguilera, 2007*) and RNase H overexpression to remove DNA:RNA hybrids (*Figure 7A*). In agreement with the role of the THO complex in R loop prevention, *hpr1Δ* caused an increase in Rad52 foci that was enhanced by AID overexpression and suppressed by RNase H overexpression, as previously reported (*Alvaro et al., 2007*; *García-Pichardo et al., 2017*; *Wellinger et al., 2006*). By contrast, the accumulation of Rad52 foci observed in *rad51Δ* cells was not affected by either AID or RNase H overexpression. This result argues that R loops are not the cause for the genetic instability observed in the absence of Rad51. The accumulation of Rad52 foci in *rad51Δ* cells is rather likely due to the accumulation of unrepaired recombination

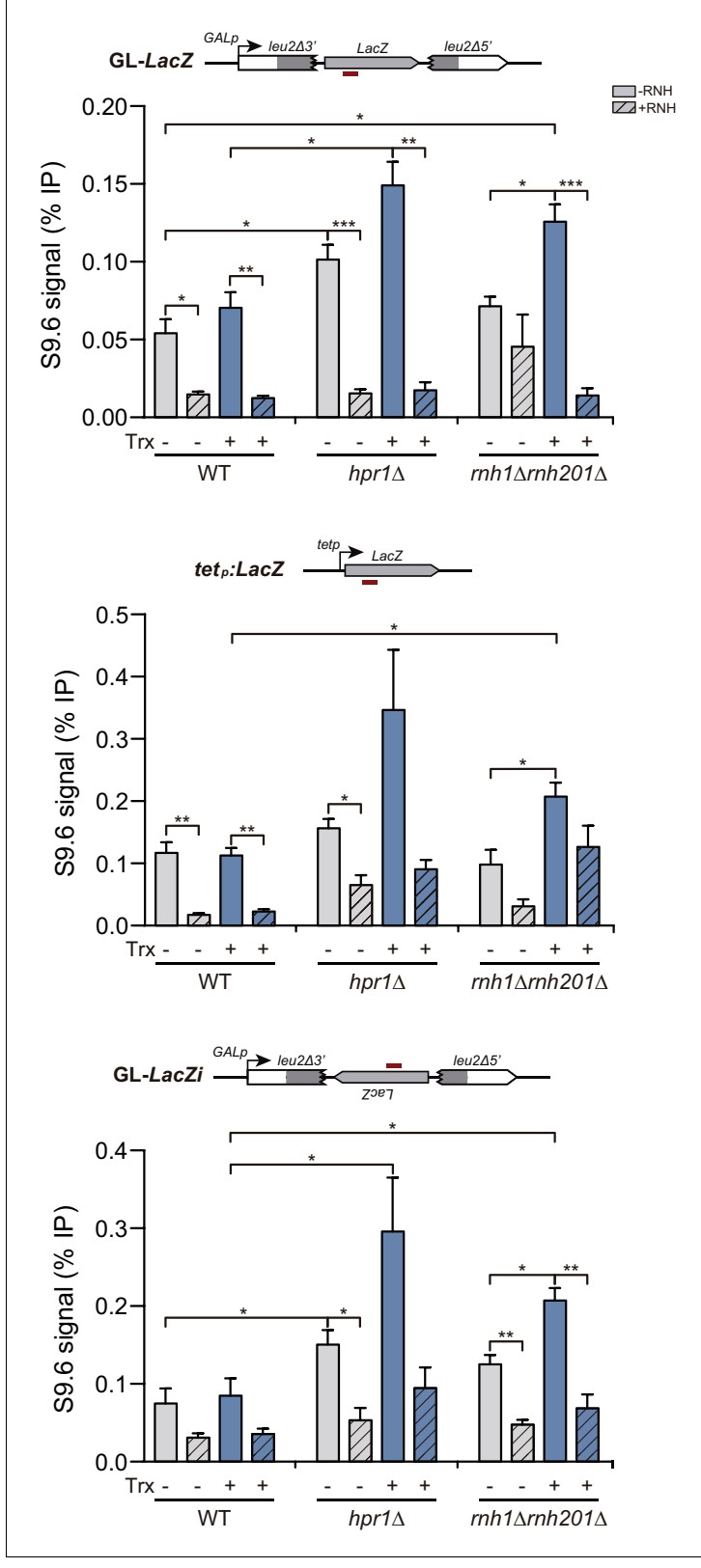

**Figure 3.** Detection of co-transcriptional DNA:RNA hybrids in *hpr1Δ* and *rnh1Δ rnh201Δ* mutants at the *LacZ*-containing constructs under the *GAL1* or *tet* promoters. DNA:RNA Immuno-Precipitation (DRIP) with the S9.6 antibody in WT (W303), *hpr1Δ* (U678.4C) and *rnh1Δ rnh201Δ* (HRN2.10C) strains in asynchronous cultures treated or not in vitro with RNase H in the GL-*LacZ*, *tetp:LacZ* and GL-*LacZi* constructs turned transcriptionally off (2%

*Figure 3 continued on next page*

*Figure 3 continued*

glucose or 5 μg/mL doxycycline) or on (2% galactose and in the absence of doxycycline). Average and SEM of three independent experiments are shown *, p≤0.05; **, p≤0.01; ***, p≤0.001 (unpaired Student's t-test). The online version of this article includes the following source data for figure 3:

**Source data 1.** Detection of co-transcriptional DNA:RNA hybrids.

intermediates, as previously suggested (*Alvaro et al., 2007*). Importantly, *hpr1Δ rad51Δ* cells showed a similar result, further supporting that the accumulation of recombinogenic damage in *hpr1Δ* cells is independent on Rad51. Consequently, we next directly measured DNA:RNA hybrid accumulation by immunodetection with the S9.6 antibody on metaphase spreads. *Figure 7B* illustrates that the number of cells with S9.6 positive signal was similar in *hpr1Δ* and in *hpr1Δ rad51Δ* cells. Altogether, these results demonstrate that the Rad51 protein is not required for the DNA:RNA hybrid formation previously reported in THO mutants.

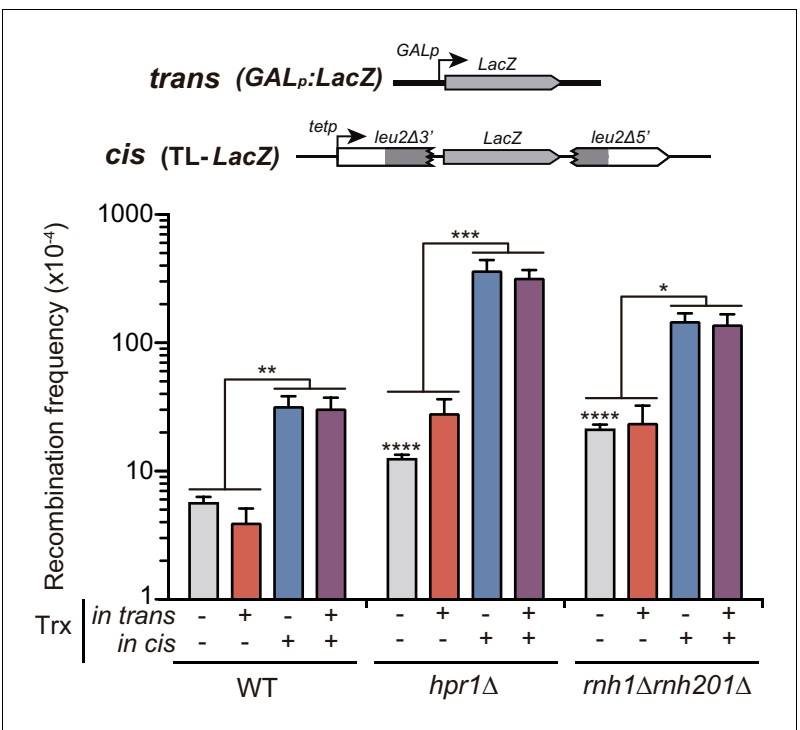

**Figure 4.** Analysis of the effect on genetic recombination of RNA produced in cis or in trans with different promoters. Recombination analysis in WT (W303), *hpr1Δ* (U678.4C) and *rnh1Δ rnh201Δ* (HRN2.10C) carrying TL-*LacZ* plasmid system (pCM184-TL-*LacZ*) plus either the pRS416 empty vector or the same vector carrying the *LacZ* gene (pRS416-*GALLacZ*). In this case, the four combinations studied were: i) no transcription, with TL-*LacZ* construct turned transcriptionally off (5 μg/mL doxycycline) and an empty plasmid; ii) transcription in trans, with TL-*LacZ* construct turned transcriptionally off (5 μg/mL doxycycline) and the *GAL-LacZ* construct switched on (2% galactose); iii) transcription in cis, with TL-*LacZ* construct turned transcriptionally on and an empty plasmid; and iv) transcription in cis and in trans, with TL-*LacZ* construct turned transcriptionally on and the *GAL-LacZ* construct switched on (2% galactose). Average and SEM of at least three independent experiments consisting in the median value of six independent colonies each are shown. *, p≤0.05; **, p≤0.01; ***, p≤0.001; ****, p≤0.0001 (unpaired Student's t-test).

The online version of this article includes the following source data for figure 4:

**Source data 1.** Analysis of the effect on genetic recombination of RNA producedin cisorin transwith different promoters.

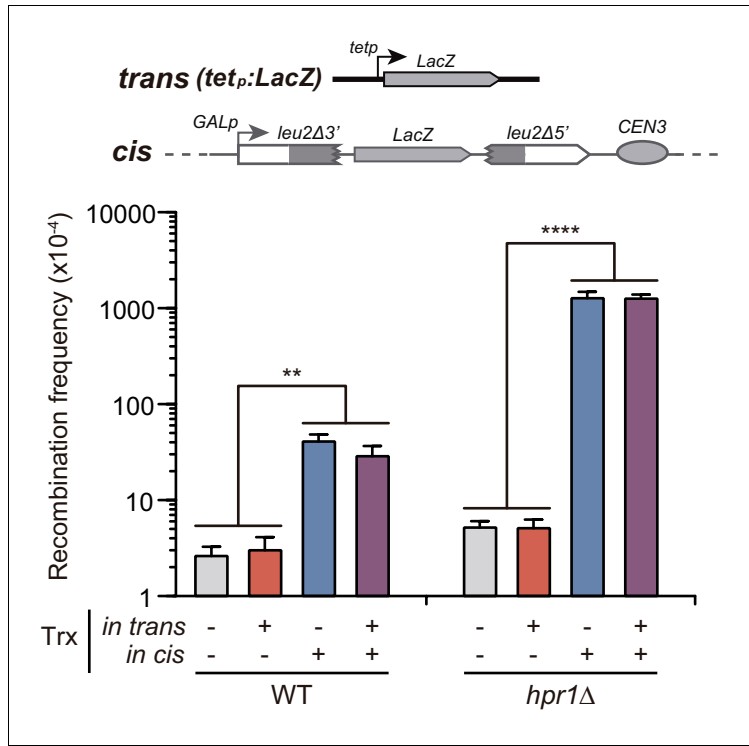

**Figure 5.** Analysis of the effect on genetic recombination of RNA produced in trans of chromosome III. Recombination analysis in WT (WGLZN), *hpr1Δ* (HGLZN) strains carrying the GL-*LacZ* recombination system integrated in chromosome III. Strains were transformed with empty vector pCM190 or the same vector carrying the *LacZ* gene (pCM179). Average and SEM of at least three independent experiments are shown consisting in the median value of six independent colonies each. **, p≤0.01; ****, p≤0.0001 (unpaired Student's t-test). The online version of this article includes the following source data for figure 5:

**Source data 1.** Analysis of the effect on genetic recombination of RNA producedin transof chromosome III.

## Discussion

We have devised a new genetic assay to infer whether the source of DNA:RNA hybrids compromising genome integrity could potentially come from RNAs produced in trans. To reach this conclusion, we used an SSA assay. It is well established that SSA events are Rad51-independent; they do not require DNA strand exchange, but just annealing between resected single-stranded DNA (ssDNA) for which the action of Rad52 is sufficient (*Figure 1A*; *Pardo et al., 2009*). Our constructs show that, in contrast to the RNA produced at the site where SSA occurs, an RNA produced in a remote locus does not induce an increase in homology-directed repair. Importantly, recombination is not induced by in trans RNA production even when the major DNA:RNA removal machinery is absent (*rnh1Δ rnh201Δ* mutant) or when the RNA coating functions preventing DNA:RNA hybrid formation are impaired (*tho* mutants), arguing against the idea that harmful DNA:RNA hybrids could spontaneously form in trans and constitute a menace for genome integrity. Co-transcriptional R-loops are responsible for the hyper-recombination of *hpr1Δ* as reported previously (*Huertas and Aguilera, 2003*). Putative DNA:RNA hybrids formed in trans would be expected to further increase recombination levels. Instead, the simultaneous induction of transcription in cis and in trans (*Figure 2A*) reduced the strong hyper-recombinogenic effect of *tho* mutants. The fact that this suppressor effect was augmented when one of the *LacZ* sequences was inverted (*Figure 2B*) and prevented by a shorter *LacZ* construct (*Figure 2C*), which was reported to be more stable in *tho* mutant backgrounds (*Chávez et al., 2001*), suggests that the free RNA itself, and not in the form of DNA:RNA hybrids formed at the template DNA strand, plays some role in preventing the hyper-recombination, likely because stable RNAs can interfere with transcription at a homologous locus. However, no suppressor effect was observed when the recombination system was placed in a chromosome (*Figure 5*) or when the ectopic RNA was transcribed from the *GAL1* promoter (*Figures 2* and *4*).

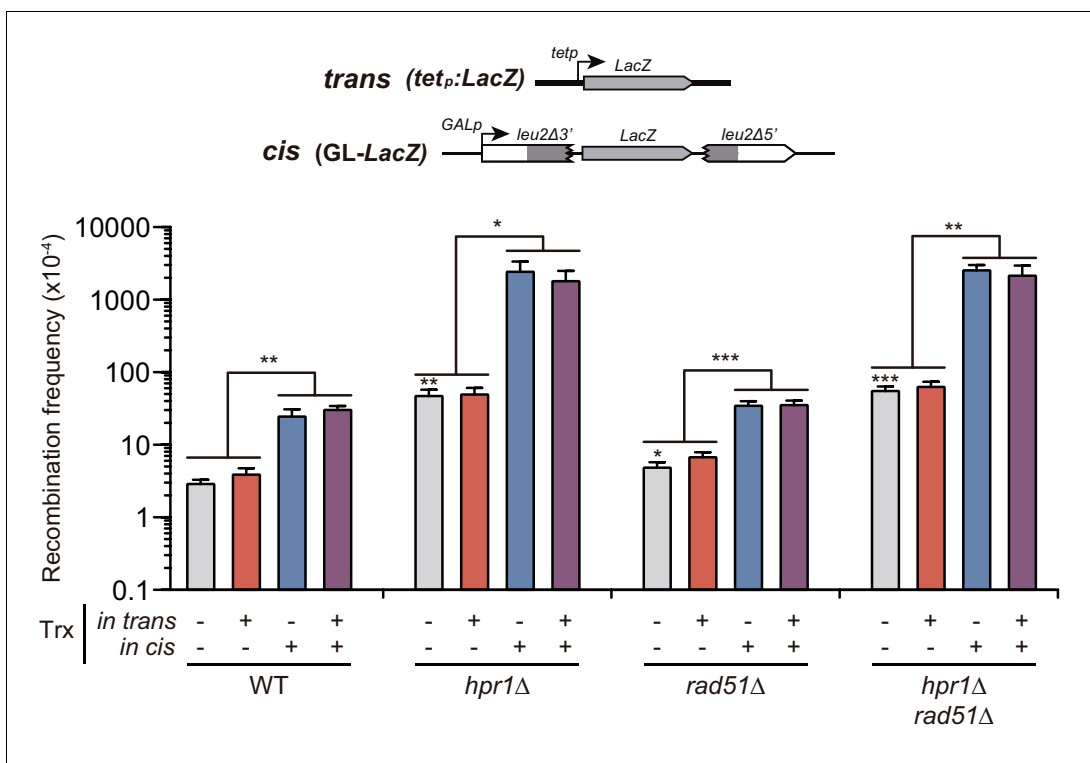

**Figure 6.** Analysis of the effect on genetic recombination of RNA produced in trans with or without Rad51. Recombination analysis in WT (W303), *hpr1Δ* (U678.1C), *rad51Δ* (WSR51.4A) and *hpr1Δ rad51Δ* (HPR51.15A) strains carrying GL-*LacZ* plasmid system (pRS314-GL-*LacZ*) plus either the pCM190 empty vector or the same vector carrying the *LacZ* gene (pCM179). Average and SEM of at least three independent experiments consisting in the median value of six independent colonies each are shown. *, p≤0.05; **, p≤0.01, ***, p≤0.001 (unpaired Student's t-test).

The online version of this article includes the following source data for figure 6:

**Source data 1.** Analysis of the effect on genetic recombination of RNA producedin transwith or without Rad51.

DNA:RNA hybrids formed with an RNA produced in trans were previously suggested to threaten genome integrity (*Wahba et al., 2013*). This conclusion was based on experiments performed with a yeast artificial chromosome after the induction of transcription of a homologous region placed at chromosome III. Recombination involving multiple substrates was first reported in *S. cerevisiae*, in which an induced-DSB triggered recombination between two other homologous fragments at different chromosomes (*Ray et al., 1989*). Tri-parental recombination assays have been successfully used since then to define specific features of the HR reaction as well as for studies of Break-Induced Recombination (BIR) or translocations and chromosomal rearrangements occurring between ectopic regions (*Pardo and Aguilera, 2012*; *Piazza et al., 2017*; *Ruiz et al., 2009*). However, such events are not the most adequate to infer recombination initiation unless this has been artificially induced (as is the case of an HO-induced DSB). Hence, the assay used to infer the potential of DNA:RNA hybrids formed with RNAs produced in trans to induce genetic instability (*Wahba et al., 2013*) relied on an RNA fragment produced at a (first) DNA region that could form a DNA:RNA hybrid with a (second) ectopic homologous DNA region that would promote its deletion or loss, leading to a genetically detectable phenotype. Thus, this assay does not exclude the possibility that the RNA forms the hybrid in cis inducing subsequently a DSB that would stimulate the recombination events studied (*Figure 6*). Indeed, this event would demand the action of Rad51 for DNA strand invasion, consistent with the results obtained (*Wahba et al., 2013*). Therefore, the increased genetic instability observed could be explained by the invasion of the 3' end of a DNA break induced by the DNA:RNA hybrid formed at the first site (*Figure 8*) rather than implying that Rad51 is required for the RNA to invade the second DNA sequence.

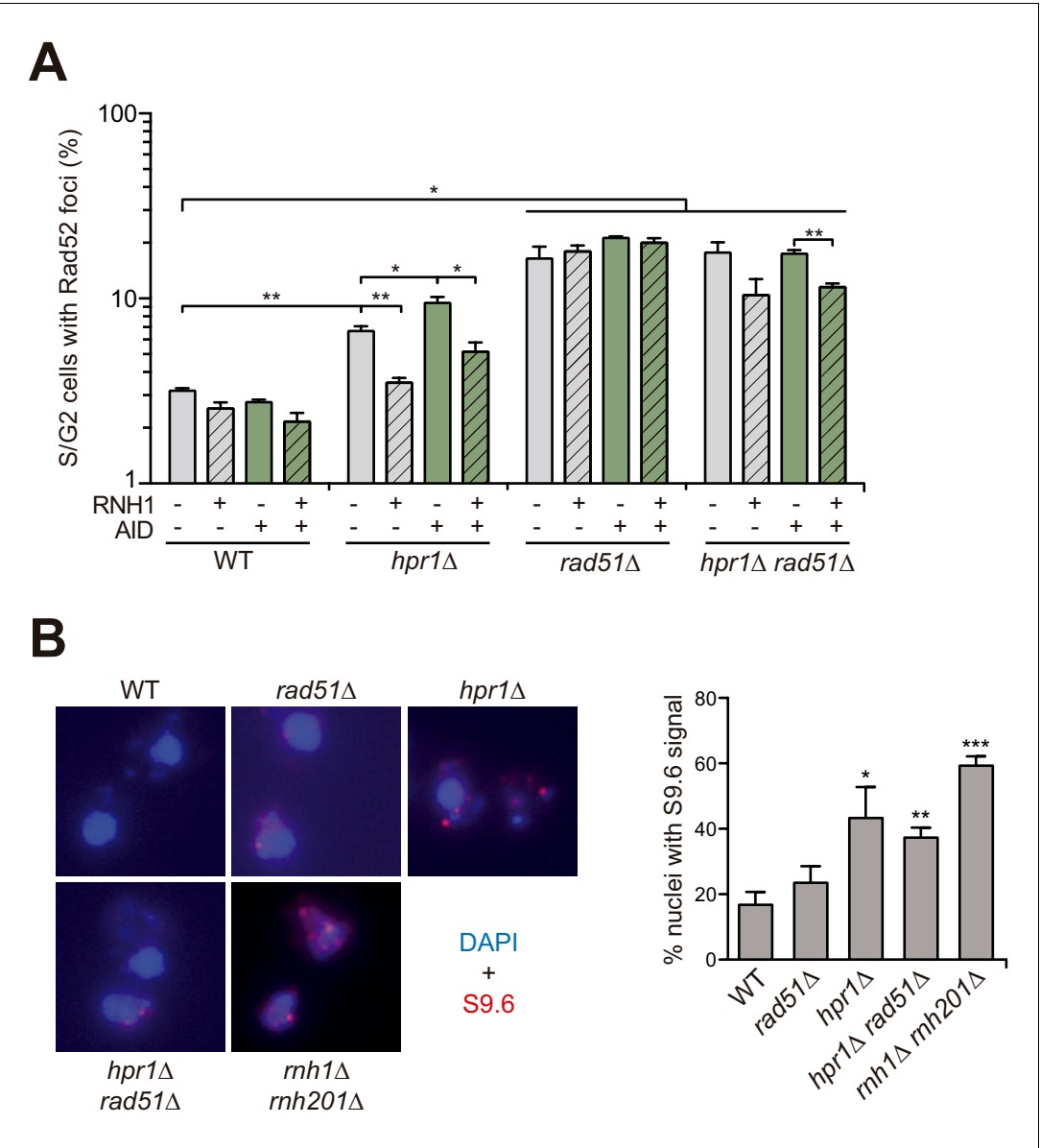

**Figure 7.** The increased genetic instability and DNA:RNA hybrids of *hpr1Δ* are independent on Rad51. (**A**) Spontaneous Rad52-YFP foci formation in WT (W303), *hpr1Δ* (U678.1C), *rad51Δ* (WSR51.4A) and *hpr1Δ rad51Δ* (HPR51.15A) strains carrying the empty vectors pCM184 and pCM189, or a combination of both carrying the RNH1 or AID genes as indicated in the legend. (**B**) Representative images and value of the percent of the total nuclei scored that stained positively for DNA:RNA hybrids in chromatin spreads stained with the S9.6 antibody in WT (W303), *hpr1Δ* (U678.1C), *rad51Δ* (WSR51.4A), *hpr1Δ rad51Δ* (HPR51.15A) and RNH-R (*rnh1Δ rnh201Δ*) strains. In both panels, average and SEM of at least three independent experiments performed with more than 100 cells are shown. *, p≤0.05; **, p≤0.01; ***, p≤0.001 (unpaired Student's t-test).

The online version of this article includes the following source data for figure 7:

**Source data 1.** Genetic instability and DNA:RNA hybrids in the absence of Rad51.

---

 In our case, however, we show that the hyper-recombinogenic potential of DNA:RNA hybrids is Rad51-independent (*Figure 4*). Our assays involve two *leu2* homologous repeats that recombine by Rad51-independent SSA. Indeed, as expected, *RAD51* deletion caused no decrease in the observed recombination frequencies in our assay (*Figure 5*). Recombination between the *leu2* repeats could be originated by either a DNA:RNA hybrid in cis or by a DSB occurring in between the repeats, or as suggested previously for *tho* mutants, by a bypass mechanism involving template switching

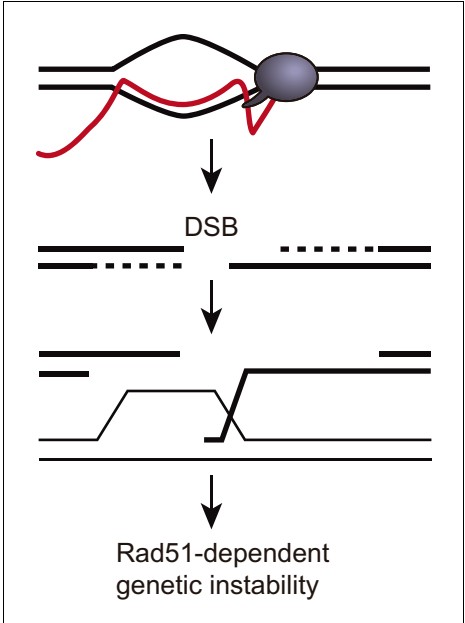

**Figure 8.** A model to explain how DNA:RNA hybrids could induce Rad51-dependent genetic instability in trans. DNA:RNA hybrid produced in cis can induce a DSB in the same sequence. The 3' end of such a DSB could invade an ectopic homologous sequence and destabilize it. This DNA strand invasion event would require Rad51. In this model, genetic instability caused by hybrids in trans would be Rad51-dependent without the need of invoking a Rad51-mediated DNA:RNA hybrid formation in trans. A DSB is depicted for simplicity, but other recombinogenic lesions such as nicks or ssDNA gaps cannot be ruled out.

(*Gómez-González et al., 2009*). It is worth noting that although we are depicting the SSA reactions as being initiated by a DSB (*Figures 1* and *8*), we cannot discard that the initial lesion triggered by a hybrid is a nick or ssDNA gap, as previously proposed for *tho* mutants (*Gómez-González et al., 2009*).

Similarly, a DSB occurring at the locus where the RNA in trans was generated could give rise to Leu+ recombinants in our assay. However, such recombination events would be Rad51-dependent, as they will require a Rad51-dependent invasion into the GL-*LacZ* construct (*Figure 8*). Hence, the Leu+ recombinants obtained in *rad51Δ* mutant cells (*Figure 5*) can only be explained by Rad51-independent events occurring in cis, at the GL-*LacZ* construct. Strikingly, the fact that we detected no significant increase in Leu+ recombinants by inducing transcription in trans, either in *RAD51* or *rad51Δ* backgrounds rules out the possibility that recombinogenic DNA:RNA hybrids form in trans in our assay. It was previously shown that S9.6 signal detected by IF was reduced by *rad51Δ* in metaphase spreads (*Wahba et al., 2013*). By contrast, we detected S9.6 signal in metaphase spreads of the *hpr1Δ* mutant of the THO complex in both *RAD51* and *rad51Δ* backgrounds (*Figure 7B*). The uncertainty about the identity of the structures detected by IF using the S9.6 antibody, which also recognizes dsRNA (*Hartono et al., 2018*; *König et al., 2017*; *Silva et al., 2018*), and the possibility that chromosomal spreads could preferentially visualize the rDNA regions, in which high levels of dsRNA structures formed by the rRNAs, makes difficult to make conclusions on S9.6 IFs in this case.

Thus, we have found no evidence for a Rad51-facilitated strand invasion from RNAs produced in trans. Further arguing against any major role of this recombinase in R loop metabolism or function, none of the so far reported DNA:RNA hybrid interactomes has identified RAD51 (*Cristini et al., 2018*; *Nadel et al., 2015*; *Wang et al., 2018*). The fact that, in vitro, RecA can catalyze an inverse DNA strand exchange reaction with DNA or RNA thus promoting the assimilation of a transcript into duplex DNA (*Kasahara et al., 2000*; *Zaitsev and Kowalczykowski, 2000*) does not argue that this is the case for unscheduled recombinogenic R loops in vivo. More likely, the biological significance of this process relies on its use for replication initiation of prokaryotic cells as originally proposed (*Zaitsev and Kowalczykowski, 2000*), for replication-dependent recombination to restart stalled forks (*Pomerantz and O'Donnell, 2008*) or even for transcription-induced origin-independent replication (*Stuckey et al., 2015*). Hence, DNA:RNA hybridization could occur in trans under regulated conditions but not spontaneously as unscheduled and harmful structures that would put genome integrity into risk. Thus, the assimilation of a transcript into a duplex DNA in trans would be tightly regulated and limited to specific reactions such as the case of telomerase or CRISPR and possibly other proteins yet to be discovered. For other cases, such as that of the GADP45 factor that binds to promoters harboring hybrids formed by lncRNAs (*Arab et al., 2019*), it is unclear whether such hybrids are formed in trans and in a GADP45-dependent manner.

Altogether, our results suggest that RNAs do not form hybrids in trans, so that the previously reported induction of Rad51-dependent ectopic genetic instability would be explained by R loop-mediated DNA breaks in cis.

# Materials and methods

## Key resources table

| Reagent type (species) or resource | Designation | Source or reference | Identifiers | Additional information |
|---|---|---|---|---|
| Genetic reagent *Saccharomyces cerevisiae* | W303 background strains with different gene deletions | various | | (See Materials and methods section) |
| Recombinant DNA reagent | Yeast expression plasmids and recombination systems | various | | (See Materials and methods section) |
| Sequence-based reagent | Primers for DRIP and RT-PCR | Condalab | | (See Materials and methods section) |
| Antibody | Cy3 conjugated anti-mouse (goat monoclonal) | Jackson laboratories | Cat# 115-165-003, RRID:AB_2338680 | IF (1:1000) |
| Antibody | S9.6 anti DNA:RNA hybrids (mouse monoclonal) | ATCC Hybridoma cell line | Cat# HB-8730, RRID:CVCL_G144 | DRIP (1 mg/ml) and IF (1:1000) |
| Commercial assay, kit | Macherey-Nagel DNA purification | Macherey- Nagel | Cat# 740588.250 | |
| Commercial assay, kit | Qiagen's RNeasy | Quiagen | Cat# 75162 | |
| Commercial assay, kit | Reverse Transcription kit | Qiagen | Cat # 205311 | |
| Peptide, recombinant protein | Zymolyase 20T | US Biological | Z1001 | (50 mg/ml) |
| Chemical compound, drug | Doxycyclin hyclate | Sigma-Aldrich | D9891 | (5 mg/ml) |
| Peptide, recombinant protein | Proteinase K (PCR grade) | Roche | Cat # 03508811103 | |
| Peptide, recombinant protein | Rnase A | Roche | Cat # 10154105103 | |
| Software, algorithm | GraphPad Prism V8.4.2 | GraphPad Software, La Jolla, CA, USA | RRID:SCR_002798 | |
| Other | iTaq Universal SYBR Green | Bio-RAD | Cat # 1725120 | |
| Other | DAPI stain | Invitrogen | D1306 | 1 µg/mL |

## Yeast strains and Plasmids

Strains used were the wild-type W303-1A (*MATa ade2-1 can1- 100 his3-11,15 leu2-3,112 trp1-1 ura3-1 rad5-G535R*) and its isogenic *hpr1∆::HIS3* mutant U678-1C (*MATa*) and U678-4C (*MATα*), *mft1∆::KANMX* mutant (WMK.1A) (*Chávez and Aguilera, 1997*), *rnh1∆::KANMX rnh201∆::KANMX* (RNH-R), *rad51∆::KANMX* (WSR51.4A) (*González-Barrera et al., 2002*), and *hpr1∆::HIS3MX rad51∆::KANMX* (HPR51.15A) from this study. *rnh1∆::KAN rnh201∆::KAN* (HRN2.8A) and the wild-type HRN2.8A were from *Huertas and Aguilera, 2003*. Wild-type (WGLZN) and *hpr1∆HIS3* mutant were made in this study by insertion of the GL-*LacZ::NATMX* at the *LEU2* locus in Chromosome III of a W303-1A strain corrected for *RAD5* (*Moriel-Carretero and Aguilera, 2010*).

Yeast plasmids pCM179, pCM184, pCM189 and pCM190 were previously published (*Garí et al., 1997*). pRS314-GL-*LacZ* (*Piruat and Aguilera, 1998*) and pRS314-GL-*LacZi* plasmids with recombination systems were built as follows. The *BamH*I fragment containing the *LacZ* sequence from pPZ (*Straka and Hörz, 1991*), was inserted in both sense and antisense orientations with respect to the

promoter, respectively, into the *Bgl*II site of pRS314GLB (*Piruat and Aguilera, 1998*). pCM184TL-lacZ (*Santos-Pereira et al., 2013*) pRS416 and pRS416-GALlacZ were described previously (*Prado et al., 1997*). pCM184:AID was built by inserting the AID ORF from pCM189:AID (*Santos-Pereira et al., 2013*) into the *Not*I site of pCM190. pCM190-tet::LacZ400 was built by cloning the *Kpn*I-*Bam*HI 400 bp fragment of the 3' from the *LacZ* gene into *Kpn*I-*Bam*HI digested pCM190. Plasmids pCM189:AID, pCM184:RNH1 (*Santos-Pereira et al., 2013*), and pWJ1344 (*Lisby et al., 2001*) were also previously published.

## Yeast transformation

Yeast transformation was performed using the lithium acetate method as previously described (*Gietz et al., 1995*).

## Recombination assays

Cells transformed were grown in selective media containing 2% glucose and 5 µg/mL of doxycycline (kept in the dark) to repress transcription from the *GAL1* and *tet* promoter, respectively. Recombination frequencies were calculated as previously described as means of at least three median frequencies obtained each from six independent colonies isolated in the appropriate medium for the selection of the required plasmids (*Gómez-González et al., 2011*). Briefly, transformants were cultured for at 3–4 days (until acquiring similar colony size) in the appropriate selective media containing either 2% glucose or 2% galactose and recombinants were obtained by plating appropriate dilutions in selective medium. To calculate total number of cells, plates with the same requirements as for the original transformation were used. All plates were grown for 3–4 days at 30°C. The average and SEM of at least three independent transformants was plotted for each figure but the numerical data can be seen in *Figure 2—source data 1*, *Figure 4—source data 1*, *Figure 5—source data 1* and *Figure 6—source data 1*.

## Transcription analysis

Mid-log cultures were grown with either glucose or galactose and with or without 5 µg/ml doxycycline (kept in the dark). Total RNA was obtained using Qiagen's RNeasy kit and used for cDNA synthesis with the QuantiTect Reverse Transcription kit with random primers (Qiagen) according to instruction. Real-time quantitative PCR was performed using iTaq universal SYBR Green (Biorad) with a 7500 Real-Time PCR machine (Applied Biosystems). Primers sequences used for this analysis were LacZT1-Fw (GCGCCGTGGCCTGAT), LacZT1-Rv (GTGCAGCGCGATCGTAATC), Intergenic-Fw (TGTTCCTTTAAGAGGTGATGGTGAT) and Intergenic-Rv (GTGCGCAGTACTTGTGAAAACC). The exact values obtained are shown in *Figure 1—figure supplement 1—source data 1*.

## DRIP assays

DNA:RNA hybrids were measured in cultures with either glucose or galactose and either with or without 5 µg/ml doxycycline (kept in the dark). Cultures were collected, washed with chilled water, resuspended in 1.4 mL spheroplasting buffer (1 M sorbitol, 10 mM EDTA pH 8, 0.1% β-mercaptoethanol, 2 mg/ml Zymoliase 20T) and incubated at 30°C for 30 min. The spheroplasts were pelleted (5 min at 7000 rpm) rinsed with water and homogeneously resuspended in 1.65 mL of buffer G2 (800 mM Guanidine HCl, 30 mM Tris-Cl pH 8, 30 mM EDTA pH 8, 5% Tween-20, 0.5% Triton X-100). Samples were treated with 40 µl 10 mg/ml RNase A for 30 min at 37°C and 75 µl of 20 mg/ml proteinase K (Roche) for 1 hr at 50°C. DRIP was performed mainly as described (*Ginno et al., 2012*) with few differences. DNA was extracted gently with chloroform:isoamyl alcohol 24:1. Precipitated DNA, washed twice with 70% EtOH, resuspended gently in TE and digested overnight with 50 U of *Hind*III, *Eco*RI, *Bsr*GI, *Xba*I and *Ssp*I, 2 mM spermidine and 2.5 µl BSA 10 mg/ml. Half of the DNA was treated with 8 µL RNase H (New England BioLabs) overnight 37°C as RNase H control. RNA-DNA hybrids were immunoprecipitated using S9.6 monoclonal antibody (hybridoma cell line HB-8730) coupled to Dynabeads Protein A (Invitrogen) for 2 hr at 4°C and washed 3 times with 1x binding buffer. DNA was eluted in 100 µL elution buffer (50 mM Tris pH 8.0, 10 mM EDTA, 0.5% SDS) treated 45 min with 7 µL proteinase K 20 mg/ml at 55°C and purified with Macherey-Nagel DNA purification kit. Primers sequences used for this analysis were LacZT1-Fw (GCGCCGTGGCCTGAT)

and LacZT1-Rv (GTGCAGCGCGATCGTAATC). The average and SEM of at least three independent transformants was plotted but the numerical data can be seen in *Figure 3—source data 1*.

## Detection of Rad52-YFP foci

Spontaneous Rad52-YFP foci from mid-log growing cells carrying plasmid pWJ1344 were visualized and counted by fluorescence microscopy in a Leica DC 350F microscope, as previously described (*Lisby et al., 2001*). More than 200 S/G2 cells where inspected for each experimental replica. The average and SEM of at least three independent transformants was plotted but the numerical data can be seen in *Figure 7—source data 1*.

## S9.6 immunofluorescence of yeast chromosome spreads

The procedure performed is similar to *Chan et al., 2014* with some modifications. Briefly, mid-log cultures (OD600 = 0.5–0.8) were grown at 30˚C; 10 ml of them were collected, washed in cold spheroplasting buffer (1.2 M sorbitol, 0.1 M potassium phosphate and 0.5 MgCl$_2$ at pH 7) and then digested by adding 10 mM DTT and 150 mg/ml of Zymolyase 20T to the same buffer. The digestion was performed for 10 min (37˚C) and stopped by mixing the samples with the solution 2 (0.1 M MES, 1M sorbitol, 1 mM EDTA, 0.5 mM MgCl$_2$, pH 6.4). Later, spheroplasts were centrifuged carefully 8 min at 800 rpm, lysed with 1% vol/vol Lipsol and fixed on slides using Fixative solution (4% paraformaldehyde/3.4% sucrose). The spreading was carried out using a glass rod and the slides were dried from 2 hr to overnight in the extraction hood.

For the immuno-staining, the slides were first washed in PBS 1X in coplin jars and then blocked in blocking buffer (5% BSA, 0.2% milk in PBS 1X) over 10 min in humid chambers. Afterwards, slides were incubated with the primary monoclonal antibody S9.6 (1 mg/ml) in a humid chamber 1 hr at 23˚ C. After washing the slides with PBS 1X for 10 min, the slides were incubated 1 hr at 23˚C in the dark with the secondary antibody Cy3 conjugated goat anti-mouse (Jackson laboratories, #115-165-003) diluted 1:1000 in blocking buffer. Finally, the slides were mounted with 50 µl of Vectashield (Vector laboratories, CA) with 1X DAPI and sealed with nail polish. For each experiment, more than 100 nuclei were visualized and counted to obtain the fraction of nuclei with DNA:RNA hybrids. The average and SEM of at least three independent transformants was plotted but the numerical data can be seen in *Figure 7—source data 1*.

## Acknowledgements

Research was supported by the Spanish Ministry of Economy and Competitiveness (BFU2016-75058-P) and the European Union (FEDER). BG-G was funded by a grant from the Spanish Association Against Cancer (AECC).

## Additional information

### Competing interests

Andrés Aguilera: Reviewing editor, *eLife*. The other authors declare that no competing interests exist.

### Funding

| Funder | Grant reference number | Author |
| --- | --- | --- |
| Ministerio de Economía y Competitividad | BFU2016-75058-P | Andrés Aguilera |
| European Regional Development Fund | | Andrés Aguilera |
| Spanish Association Against Cancer | | Belén Gómez-González |

The funders had no role in study design, data collection and interpretation, or the decision to submit the work for publication.

## Author contributions
Juan Lafuente-Barquero, Conceptualization, Data curation, Formal analysis, Validation, Writing - review and editing; Maria Luisa García-Rubio, Formal analysis, Investigation, Methodology; Marta San Martin-Alonso, Formal analysis; Belén Gómez-González, Conceptualization, Formal analysis, Supervision, Funding acquisition, Writing - original draft, Writing - review and editing; Andrés Aguilera, Conceptualization, Supervision, Funding acquisition, Writing - original draft, Writing - review and editing

## Author ORCIDs
Belén Gómez-González (iD) https://orcid.org/0000-0003-1655-8407
Andrés Aguilera (iD) https://orcid.org/0000-0003-4782-1714

## Decision letter and Author response
Decision letter https://doi.org/10.7554/eLife.56674.sa1
Author response https://doi.org/10.7554/eLife.56674.sa2

# Additional files

## Supplementary files
• Transparent reporting form

## Data availability
All data generated or analysed during this study are included in the manuscript and supporting files. Source data files have been provided for all graphs.

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
