## [Decision Letter]

**Acceptance summary:**

Using a clever genetic system in the budding yeast *Saccharomyces cerevisiae* the authors test whether R-loops can form in trans, meaning that a transcript from locus A could lead to R-loop formation in locus B. Moreover, they test whether R-loop formation is dependent on Rad51, the eukaryotic RecA family recombinase. Using their genetic system and the S9.6 antibody to detect R-loops in wild type and strains with mutations known to affect R-loop, conclusive data are shown that R-loops only form in cis and that R-loops in this genetic system are independent of Rad51. Overall, this work significantly enriches the discussion in the R-loop field and provides an alternative view point of an earlier publication that suggested R-loop formation in trans being catalyzed by Rad51.

**Decision letter after peer review:**

Thank you for submitting your article "Harmful DNA:RNA hybrids are formed in cis and in a Rad51-independent manner" for consideration by *eLife*. Your article has been reviewed by three peer reviewers, including Wolf-Dietrich Heyer as the Reviewing Editor and Reviewer #1, and the evaluation has been overseen by Jessica Tyler as the Senior Editor. The following individual involved in review of your submission has agreed to reveal their identity: Catherine H Freudenreich (Reviewer #2).

The reviewers have discussed the reviews with one another and the Reviewing Editor has drafted this decision to help you prepare a revised submission.

As the editors have judged that your manuscript is of interest, but as described below that additional experiments are required before it is published, we would like to draw your attention to changes in our revision policy that we have made in response to COVID-19 (https://elifesciences.org/articles/57162). First, because many researchers have temporarily lost access to the labs, we will give authors as much time as they need to submit revised manuscripts. We are also offering, if you choose, to post the manuscript to bioRxiv (if it is not already there) along with this decision letter and a formal designation that the manuscript is 'in revision at *eLife*'. Please let us know if you would like to pursue this option. (If your work is more suitable for medRxiv, you will need to post the preprint yourself, as the mechanisms for us to do so are still in development.)

Summary:

Using a clever genetic system in the budding yeast *Saccharomyces cerevisiae* the authors test whether R-loop can form in trans, meaning that a transcript from locus A could lead to R-loop formation in locus B. Moreover, they test whether R-loop formation is dependent on Rad51, the eukaryotic RecA family recombinase. Using their genetic system and cytological analysis of Rad52 foci and the S9.6 antibody to detect R-loops in wild type and strains with mutations known to affect R-loop, conclusive data are shown that R-loops only form in cis and that R-loops in this genetic system are independent of Rad51. Overall, this work significantly enriches the discussion in the R-loop field and provides an alternative view point of an earlier publication that suggested R-loop formation in trans being catalyzed by Rad51.

Essential revisions:

1) The pGal promoter induces very high transcription in the presence of galactose (often 500X or more induction). The level is likely very different (much less) for the tet promoter, which is generally only induced 2-3X upon addition of doxycycline. This could significantly affect the results – e.g. the cis vs. trans effects could really be a matter of different transcription levels. Transcription levels from each promoter really need to be determined- this is a very important control. The exact induction conditions used, including concentrations and induction times, need to be spelled out in the Materials and methods and should be consistent with those used during the RT-PCR experiment to test transcript levels. In the absence of being able to do the experiment on the constructs used (which would be optimal), at least it could be cited if this lab has used the same promoters and induction conditions in the past, and a caveat inserted if transcription levels are different. It would also be good to switch the promoters and make sure the result holds, as there could be issues of differences in timing of transcription as well.

2) In Figure 2 the authors relate recombination frequencies in their assays to RNA:DNA hybrid formation without measuring hybrids directly. This is a major weakness that significantly limits data interpretation. For instance, I am very surprised that the "cis" recombination frequency of the inverted *LacZ* reporter is essentially as high as the regular *lacZ* construct. This result implies that hybrid formation is insensitive to the orientation of the reporter when in many reported cases, R-loop formation is strongly orientation-dependent. Of course, another hypothesis is that (stalled) transcription itself triggers recombination, not R-loops. Without data on R-loop formation, one cannot disentangle transcription from co-transcriptional R-loop formation. The authors must use DRIP-based assay to quantify R-loop levels in the various sequence contexts and under the various genetic backgrounds to establish that their assay is reflective of R-loop levels. Using bisulfite-based readouts to measure R-loop distributions and lengths across the *LacZ* region would be even better. Without this data, the claim that this new genetic assay can "infer the formation of recombinogenic DNA:RNA hybrids" is unsubstantiated.

3) Source data:

The source data file should be labeled better. Missing are:

-what the numbers in the table are (rates of Leu+ x 10^-4?)

-which data goes with which figure panel

-average and SEM numbers should be shown in the data table

The exact p values not reported and could be added to source data file.

N values can be discerned from the source data file but it would be nice for them to be stated in the figure legends.

---

## [Author Response]

Essential revisions:1) The pGal promoter induces very high transcription in the presence of galactose (often 500X or more induction). The level is likely very different (much less) for the tet promoter, which is generally only induced 2-3X upon addition of doxycycline. This could significantly affect the results – e.g. the cis vs. trans effects could really be a matter of different transcription levels. Transcription levels from each promoter really need to be determined- this is a very important control. The exact induction conditions used, including concentrations and induction times, need to be spelled out in the Materials and methods and should be consistent with those used during the RT-PCR experiment to test transcript levels. In the absence of being able to do the experiment on the constructs used (which would be optimal), at least it could be cited if this lab has used the same promoters and induction conditions in the past, and a caveat inserted if transcription levels are different. It would also be good to switch the promoters and make sure the result holds, as there could be issues of differences in timing of transcription as well.

Thank you for this comment. We have addressed this issue by switching the promoters (new Figure 4) and, indeed, the result holds. In addition, we have checked the transcription levels by RT-PCR (new Figure 1—figure supplement 1) and indicated the exact time and conditions of induction of transcription in the Materials and methods section for further support.

As a consequence, it now reads:

‘Given that the levels of transcription from the GAL and tet promoters, used for the constructs are very different (Figure 1—figure supplement 1), we decided to perform recombination tests with similar constructs in which the promoters were switched. […] Figure 4 shows that, whereas transcription at the TL-*LacZ* recombination system enhanced recombination as previously published (Santos-Pereira et al., 2013), again no significant stimulation of recombination was detected when RNAs were produced in trans in either wild-type, *hpr1*∆ or *rnh1*∆ *rnh201*∆ cells even when the RNA was generated from the strong *GAL1* promoter.’

2) In Figure 2 the authors relate recombination frequencies in their assays to RNA:DNA hybrid formation without measuring hybrids directly. This is a major weakness that significantly limits data interpretation. For instance, I am very surprised that the "cis" recombination frequency of the inverted LacZ reporter is essentially as high as the regular lacZ construct. This result implies that hybrid formation is insensitive to the orientation of the reporter when in many reported cases, R-loop formation is strongly orientation-dependent. Of course, another hypothesis is that (stalled) transcription itself triggers recombination, not R-loops. Without data on R-loop formation, one cannot disentangle transcription from co-transcriptional R-loop formation. The authors must use DRIP-based assay to quantify R-loop levels in the various sequence contexts and under the various genetic backgrounds to establish that their assay is reflective of R-loop levels. Using bisulfite-based readouts to measure R-loop distributions and lengths across the LacZ region would be even better. Without this data, the claim that this new genetic assay can "infer the formation of recombinogenic DNA:RNA hybrids" is unsubstantiated.

We have determined the RNA:DNA hybrids at the various sequence contexts by DRIP as suggested. Results are shown in new Figure 3. We have observed that DNA:RNA hybrids accumulate in both, *hpr1∆* and *rnh1∆ rnh201∆* mutants as expected.

On the other hand, the hyper-recombination of *hpr1∆* when the transcribed sequence is inverted has been previously reported. In those previous studies we showed that the length (and the GC content) and not the orientation of the sequence impairs transcription and triggers hyper-recombination (Chavez et al., 2001). This is now remarked on as follows:

‘We detected a strong hyper-recombination in *hpr1*∆ cells when the *LacZ* sequence was transcribed in agreement with previous reports and with the fact that it has been shown that it is the length (and the GC content) but not the orientation of the *lacZ* sequence what impairs transcription and triggers hyper-recombination (Chavez and Aguilera, 1997; Chavez et al., 2001).’

Thus, R-loop formation was also expected in the inverted sequence as indeed it is shown in the new results of Figure 3. Perhaps we did not explain ourselves properly. In this study we do not pursue to show that co-transcriptional R-loops are responsible for the hyper-recombination of *hpr1∆* since, which was reported previously (Huertas et al., 2003), as now stated in the Discussion. We aim to probe that RNAs do not have any recombinogenic potential when produced from an ectopic locus, in trans, in contrast to RNAs produced in cis. We hope we have now clarified the message.

3) Source data:The source data file should be labeled better. Missing are:-what the numbers in the table are (rates of Leu+ x 10^-4?)-which data goes with which figure panel-average and SEM numbers should be shown in the data tableThe exact p values not reported and could be added to source data file.N values can be discerned from the source data file but it would be nice for them to be stated in the figure legends.

Performed as requested. The source data files now include average and SEM values as well as all the both the exact p values and the asterisks for the statistics depicted in all the graphs. Thank you.